# Recent Progress of Nanomedicine for the Synergetic Treatment of Radiotherapy (RT) and Photothermal Treatment (PTT)

**DOI:** 10.3390/cancers17142295

**Published:** 2025-07-10

**Authors:** Maria-Eleni Zachou, Ellas Spyratou, Nefeli Lagopati, Kalliopi Platoni, Efstathios P. Efstathopoulos

**Affiliations:** 1Department of Applied Medical Physics, Medical School, Attikon University Hospital, National and Kapodistrian University of Athens, 11527 Athens, Greece; zachoumar@gmail.com (M.-E.Z.); spyratouellas@gmail.com (E.S.); polaplatoni@gmail.com (K.P.); 2Laboratory of Biology, Department of Basic Medical Sciences, Medical School, National and Kapodistrian University of Athens, 11527 Athens, Greece; nlagopati@med.uoa.gr

**Keywords:** metal nanoparticles, nano-oncology, radiotherapy, photothermal therapy

## Abstract

Treating cancer effectively while protecting healthy tissue remains a major challenge in oncology. This review focuses on a promising strategy that combines two therapies: radiotherapy, which damages tumor DNA using ionizing radiation, and photothermal therapy, which uses near-infrared light to heat and destroy cancer cells. Both therapies are enhanced by specially designed nanoparticles that accumulate in tumors, increasing precision and reducing side effects. These nanoparticles can amplify radiation damage, generate localized heat, interfere with cancer cell repair mechanisms, and even stimulate immune responses. By analyzing recent studies, we classify the most effective nanoparticle materials—such as gold, bismuth, and copper—and explore their dual role in treatment and real-time imaging. This combined approach offers improved tumor targeting, fewer side effects, and the potential for personalized cancer therapy. Our review highlights how nanotechnology is driving innovation in cancer treatment and identifies key challenges that must be addressed before these methods reach clinical use.

## 1. Introduction

Despite tremendous advances in modern medicine, cancer remains one of the major problems for human health [1]. The most prominent treatment methods for the control of the disease are surgery, radiotherapy, chemotherapy, and immunotherapy [2]. In each one of these fields there is ongoing development, and new breakthroughs come to assist the battle against the disease. Nanomedicine, the use of nanotechnology in medicine, has been implemented in oncology with exceptional results [3]. NPs have been used as drug delivery agents in chemotherapy and immunotherapy, and as radiosensitizers and thermosensitizers in radiotherapy and hyperthermia, enhancing the treatment results [4,5]. A fascinating new approach is the utilization of dual-purpose NPs to combine different treatment methods with complementary results [6].

The integration of nanotechnology into oncology, often referred to as nano-oncology, represents a transformative approach to cancer diagnosis and treatment. As outlined by Thakor and Gambhir, nano-oncology leverages engineered nanoscale materials—typically between 1 and 100 nm—for targeted drug delivery, improved imaging, and site-specific therapy, with the aim of overcoming the limitations of conventional cancer modalities [7].

Nanoparticles offer multiple advantages in cancer treatment and diagnosis, owing to their physicochemical versatility and ability to be functionalized for targeted delivery. Their nanoscale size allows them to exploit the enhanced permeability and retention (EPR) effect, passively accumulating at tumor sites while minimizing uptake by healthy tissues [8]. Moreover, surface modification with targeting ligands (e.g., antibodies, peptides, or folic acid) enables active targeting of cancer cells, further improving their specificity [9]. Some nanoparticles can be guided to tumors using external magnetic fields (e.g., magnetic nanoparticles), allowing site-specific delivery and minimizing systemic toxicity [10]. In therapy, magnetic hyperthermia—where magnetic NPs generate localized heat under an alternating magnetic field—can selectively damage cancer cells without harming surrounding tissues [11]. NPs can also be designed for pH-sensitive drug release, ensuring that therapeutic agents are released preferentially in the acidic tumor microenvironment, reducing off-target effects [12].

From a drug delivery perspective, nanoparticles offer high drug-loading capacity, controlled release profiles, and co-delivery of multiple agents (e.g., chemotherapeutics and radiosensitizers), enhancing treatment efficacy through multimodal mechanisms [13]. In diagnosis, NPs serve as powerful contrast agents for imaging techniques such as magnetic resonance imaging (MRI), computed tomography (CT) imaging, photoacoustic imaging (PAI), and near-infrared fluorescence (NIRF) imaging [14], improving the resolution and sensitivity of tumor detection [15,16]. Altogether, these properties establish nanoparticles as a cornerstone of modern theranostics, enabling integrated cancer diagnosis, targeted therapy, and real-time monitoring.

In this context, the combination of nanoparticle-mediated radiotherapy (RT) and photothermal therapy (PTT) exemplifies the theranostic potential of nano-oncology [17]. Such multifunctional NPs are capable of enhancing radiation-induced cytotoxicity while also converting near-infrared (NIR) light into localized heat, leading to a controlled temperature rise and subsequent thermal damage, thereby achieving a synergistic therapeutic effect [18]. Furthermore, these platforms often enable real-time tumor imaging (e.g., via CT, MRI, or PET), facilitating image-guided treatment and personalized care [19]. NP-mediated RT–PTT strategies therefore represent a promising frontier within nano-oncology, offering enhanced specificity, dual-modality treatment, and improved therapeutic outcomes.

This review serves as a timely and focused synthesis of the current progress in the development of nanoparticle-based platforms for the combined application of radiotherapy and photothermal therapy. While existing reviews address RT or PTT individually [20,21], to our knowledge, they have not critically examined their combinational integration at the nanoscale. Given the growing interest in multifunctional nanomedicine and the push toward image-guided and personalized cancer treatments, this review provides researchers and clinicians with an up-to-date resource to understand the material design strategies, biological mechanisms, and translational challenges specific to NP-mediated RT–PTT. By highlighting recent preclinical advances and comparing material systems, this review aims to support future research directions and inform rational design for next-generation cancer nanotherapeutics.

### 1.1. Phototherapy and Photothermal Agents

Photothermal therapy is a method that utilizes visible light sources—mostly near-infrared (NIR) lasers—and light-absorbing agents to induce a local temperature rise inside the tumor. The purpose of the light-absorbing agents is the conversion of light energy to heat, which can induce thermal damage to cancer cells. The rise in temperature ranges between 42 and 45 °C. This temperature is high enough to harm cancer cells while sparing healthy tissue [22]. Temperatures above 45 °C cause thermal ablation [23]. The rise of temperature is caused by the mediation of the so-called PTT agents. PTT agents act as energy converters. When they get irradiated with a specific wavelength, they absorb light energy and jump from their ground state to an excited singlet state [24]. Following this, the excited PTT agents interact with their surrounding molecules, and they lose energy and return to their ground state via nonradiative vibrational relaxation. The energy loss of relaxation is transferred to the environment in the form of heat [25].

The mechanism of action behind heat-induced cell damage can be either heat shock or apoptosis by the reprogramming of the cellular cycle [26]. The results of the treatment depend on various factors, such as the temperature rise, duration of exposure, and type of tissue, but are always based on one principle, the sensitivity of proteins to heat. Cellular proteins are affected by the temperature rise of the environment by unfolding and thus losing their functionality [27]. Furthermore, the rise of temperature signals the production of heat shock proteins (HSPs) that will cause further damage to cellular proteins [28]. Nucleic proteins are the most sensitive to heating, and become aggregated inside the nucleus, while simultaneously, heat induces an imbalance between the production and metabolization of reactive oxygen species, leading to increased oxidative stress. In these ways, a temperature rise can induce DNA damage, and at the same time, heating interferes with DNA repair mechanisms, such as the repair of faulty bases, leading to permanent harm [29]. In fact, there are several different DNA repair pathways with which a temperature rise interferes, making heat an appropriate candidate for synergistic therapies [30].

It is already known that that cell death caused by heat induction can lead to an immunogenic response, but a more recent find dictates that the apoptotic cell death caused by a temperature rise can in fact be held as immunogenic cell death (ICD) [31]. Hyperthermia causes endoplasmic reticulum (ER) stress; it generates ROS and releases key damage-associated molecular patterns (DAMPs). This process leads to a combined effect of direct killing of the tumor and long-term immune activation [32].

NPs have shown great promise as PTT agents due to their unique characteristics [33]. Metal NPs, such as those composed of gold (Au), silver (Ag), copper (Cu), platinum (Pt), and palladium (Pd), exhibit the unique phenomenon of localized surface plasmon resonance (LSPR) [34]. Upon exposure to light, conduction band electrons on the NP surface begin to oscillate coherently with the incident electromagnetic field [35]. These plasmonic oscillations generate heat, which is subsequently transferred to the surrounding tissue. The oscillation reaches its maximum amplitude at a specific frequency known as the LSPR frequency [36]. In photothermal therapy (PTT), when the laser wavelength matches the LSPR frequency, heat generation becomes significantly more efficient [37]. These nanoparticles are typically synthesized in metallic (bulk-phase) form, but they can also appear as metal oxides (e.g., CuO or PdO) or composites combining a plasmonic core with a dielectric or polymer shell to improve biocompatibility and optical tunability [38]. Ranging from 10 to 100 nm in size, these NPs tend to accumulate preferentially in tumor tissue due to the enhanced permeability and retention (EPR) effect. As a result, the temperature rise is highly localized to the tumor site, minimizing damage to surrounding healthy tissues [39]. These properties make plasmonic NP-based PTT a promising and effective standalone modality for cancer treatment.

### 1.2. Radiotherapy and Radiosensitizers

Radiotherapy, along with surgery, is the most common cancer treatment. At least 50% to 70% of cancer patients will receive radiotherapy, either alone, or in combination with other treatment schemes like surgery and chemotherapy [40]. In the last decade, tremendous advances have been made in the field of RT. New technologies have led to new treatment delivery methods, like Volumetric Modulated Arc Therapy (VMAT) and radiosurgery, that are patient-specific and allow for extremely localized dose deposition [41,42]. However, these techniques are not always applicable and, in most cases, the main problem for RT remains; it is not possible to deliver a dose high enough for tumor control, while at the same time sparing the healthy surroundings [43].

There are two main mechanisms of action in radiotherapy. The first one relies on the direct interaction of ionized particles and the DNA of the cells causing double-strand breaks (DSBs) and single-strand breaks (SSBs) [44]. The second one is based on the creation of reactive oxygen species (ROS) that will chemically affect DNA molecules and cause apoptosis in cancer cells [45]. The presence of oxygen is very important in RT, as it is vital for the development of ROS. However, the unregulated growth of tumors leads to the development of an insufficient vasculature, and thus poor tumor oxygenation. To boost the effect of radiotherapy, radiosensitizers have been implemented [46,47,48].

In the last decades, various studies have been conducted on high atomic number (Z) metallic NPs that were proven to have a significant radiosensitization effect [49,50]. This effect arises from their high atomic number, which enhances the probability of photoelectric interactions (~Z^3^) within the tumor, thereby leading to increased local radiation dose deposition. Among the most prominent high-Z metallic NPs studied are gold (Au), gadolinium (Gd), hafnium (Hf), bismuth (Bi), and platinum [51]. Among them, gold (Au) and gadolinium (Gd) nanoparticles are the most extensively studied, offering both physical dose enhancement and biological effects such as ROS generation and DNA damage. Hafnium oxide (HfO_2_) has shown promising clinical translation through the NBTXR3 formulation in trials [52,53,54]. NPs induce a low systemic toxicity and can have targeted action, as they allow for selective deposition inside the tumor via the enhanced permeability and retention effect (EPR), as well as through their functionalization with tumor-specific ligands [55,56]. Furthermore, their high surface-area-to-volume ratio can facilitate the loading of drugs and other therapeutic agents, like peptides, antibodies, etc., making them suitable for multimodal treatments [57].

### 1.3. Synergistic Action of RT and PTT

In recent years, numerous studies have demonstrated that combining nanoparticle-mediated radiotherapy (RT) and photothermal therapy (PTT) yields a complementary and synergistic therapeutic effect [58]. Multifunctional nanoparticles (NPs) can act as both efficient photothermal transducers and radiosensitizers, enhancing radiation-induced DNA damage and oxidative stress. The localized hyperthermia induced by PTT not only improves tumor perfusion and oxygenation—thereby alleviating hypoxia and overcoming radioresistance—but also disrupts DNA repair mechanisms and denatures cellular proteins, further sensitizing tumor cells to ionizing radiation [59].

Compared to other multimodal treatments such as RT combined with chemotherapy or immunotherapy, NP-mediated RT–PTT offers superior spatial control and reduced systemic toxicity. Traditional chemoradiation approaches often suffer from off-target effects, poor tumor selectivity, and treatment resistance, especially in solid tumors [60]. In contrast, the dual action of RT–PTT confines both radiation enhancement and thermal damage to the tumor site, enabling effective tumor control even at lower radiation doses. Moreover, some RT–PTT platforms can be integrated with real-time imaging and may elicit immunogenic responses, offering potential for future triple-modality strategies. Together, these features position NP-mediated RT–PTT as a powerful, targeted, and more controllable alternative to conventional combination therapies

## 2. The Use of Metallic NPs for the Synergetic Effect of RT and PTT

### 2.1. Gold NPs (GNPs)

GNPs are widely used in various therapeutic and diagnostic applications due to their unique properties. Their high Z number offers a strong photoelectric absorption coefficient, they are biocompatible with low biological toxicity, and their well-controlled size distribution, in the range of 1–150 nm, offers them distinctive optical, electrical, and chemical properties. Additionally, their high surface-area-to-volume ratio renders them ideal for loading drugs and other therapeutic agents [61]. GNPs are considered as plasmonic NPs, presenting strong light absorption due to LSPR, as described above [62]. For that reason, gold NPs are widely used for photothermal applications. The synergy of the two different treatment modalities has been investigated in various studies.

In an in vitro study conducted by Neshastehriz et al., folate-conjugated spherical GNPs (F-GNPs) (5–20 nm) were used in KB mouth epidermal carcinoma cells [63]. Folate is essential for DNA synthesis and thus, it is a promising targeting ligand for directing GNPs to cancer cells, particularly in head-and-neck tumors. A cell viability assay conducted with MTT confirmed the biocompatibility of F-GNPs when used alone. A combination of 40 g/mL NPs with laser irradiation (532 nm, 0.47 W/cm^2^, 15 min) led to significant cell death (68% cell viability). Additionally, the combination of F-AuNPs with 2 Gy of 6 MV X-ray reduced cell viability by 35%, with a Dose Enhancement Factor (DEF) of 1.46. In combined RT + PTT, cell viability was further decreased to 54%, and flow cytometry with annexin V staining showed 27.76% apoptosis, considerably more significant than in the other treatment groups.

In another study, Zhang et al. studied the results of acid-induced aggregated GNPs + RT + PTT on a MCF-7 breast cancer mouse model [64]. The strategy of acid-induced aggregation was developed in a previous study of the scientific team [65]. They manufactured an AuNP system that consisted of two kinds of 30 nm GNPs, one modified with an Asp-Asp-Asp-Asp-Asp-Cys peptide, and the other with a 2,3-Dimethylmaleic anhydride (DA)-grafted Lys-Gly-Gly-Lys-Gly-Gly-Lys-Cys peptide, both with a negative surface charge. The NPs were named GNPs-A and GNPs-B, respectively. After reaching the tumor’s acidic environment, the negative surface charge of GNPs-B changed to positive, facilitating electrostatic interactions with GNPs-A, creating larger GNP aggregates. The larger size of these aggregates led to enhanced retention at the tumor site. The nanosystem significantly enhanced RT effects, with a Sensitization Enhancement Ratio (SER10) of 1.52 for RT + GNPs and 1.68 for RT + GNPs + PTT, compared to the lower SER values reported for single, non-aggregated AuNPs in previous studies. Furthermore, RT enhancement was also observed in MCF-7 xenografts, while the photoacoustic (PA) signal was also highly intensified.

Additionally, larger aggregates absorb more NIR, enhancing the photothermal conversion efficiency (PCE), thus making the NP system ideal for RT + PTT combinational therapy. For the experiments, MCF-7 breast cancer cells were injected subcutaneously into BALB/c nude mice, which were treated with different combinations of GNPs, 4 Gy RT, and PTT. For PTT treatment, an 808 nm laser (2.0 W/cm^2^, 3 min) was applied in vivo when the tumor size reached 100 mm^3^. Additional photothermal studies showed that GNP aggregates could heat tumor tissues up to 52 °C after 8 min of irradiation, though GNPs alone had minimal impact on tumor growth, while radiotherapy (RT) induced moderate suppression but allowed regrowth. PTT partially inhibited tumors (5~2 °C) but was insufficient for complete ablation.

Combining GNPs + RT’s enhanced tumor suppression, GNPs + RT + PTT achieved the most significant tumor regression with no recurrence for 20 days. Hematoxylin and eosin (H&E) staining and TUNEL assays confirmed increased apoptosis and necrosis in treated tumors, with minimal damage to surrounding healthy tissues. Biocompatibility studies showed no significant weight loss, no major organ toxicity, and no hemolysis, confirming the safety and selectivity of the treatment.

Alginate (Alg) is a natural and biocompatible polymer that is implemented for the green synthesis and stabilization of gold NPs, alleviating the process from toxic reducing agents [66]. However, its most prominent drawback is the formation of a wide size distribution of NPs. The solution to this problem arises from the conjugation of Alg with dopamine (DA) molecules. The final Alg-Da product ensures the formation of monodisperse, stable, green, and biocompatible GNPs (G@Alg-DA NPs).

In a recent study conducted by Ghaffarlou et al., the synergistic effect of RT + PT + NPs on 4T1 tumors was studied with the use of alginate-coated AuNPs [67]. In this study, the experiments consisted of both in vitro and in vivo settings. For the in vitro experiments, 4T1 tumor cells were cultivated, and PTT was applied with an NIR 1 W/cm^2^ laser for 5 min. For RT, 4 Gy of 6 MV X-rays were delivered. The various assays that were conducted 24 h post-irradiation yielded some very important results (Figure 1). Particularly, the MTT assay results showed that Au@Alg-DA NPs + RT + PTT led to a significant reduction in viability (~35%) compared to RT + NPs and NPs + PTT. The measurement of ROS took place with the DCFH-DA (2′,7′-dichlorodihydrofluorescein diacetate) fluorescent probe, which detects oxidative stress within cells. While X-ray irradiation alone induced some ROS production, the synergistic effect of PTT + RT led to a substantial increase in oxidative stress, enhancing cancer cell damage. Those results were also validated by the colony formation assay (lowest colony formation, with a survival fraction of 0.37), and Calcein-AM/PI staining, where the triple combination treatment resulted in the highest red fluorescence (dead cells) and lowest green fluorescence (live cells).

For the in vivo experiment, 1 × 10^4^ T1 cells were implanted subcutaneously in Balb/C mice. The treatment was initiated when the tumors reached a volume of 280 mm^3^. The triple combination was once again the most effective treatment option, leading to substantial tumor growth inhibition (Figure 2). Unlike the in vitro treatment, the laser irradiation was only 2 min per session. During the treatment period, there were no significant changes in body weight, indicating that the treatment was safe and biocompatible. Furthermore, the histological analysis of the tumors showed large necrotic areas and reduced cellularity in the triple combination group, while major organs like the heart, liver, kidney, and spleen displayed no significant histological damage. Tumor suppression was most significant in the triple combination group, with no recurrence for 21 days.

In a unique approach, a study conducted by Zuo et al. combined not only the effects of RT and PTT but extended the synergistic therapy with the implementation of gas therapy [68]. More specifically, the team developed a nanoplatform consisting of gold nanocages (GNCs) loaded with thiolate cupferron, a hyperpyrexia-sensitive nitric oxide (NO) donor, resulting in the final form of GNCs@NO. The NO gas which is released when hyperpyrexia (temperature > 41.5 °C) is induced reacts with ROS that are produced from RT and forms the more lethal reactive nitrogen species (RNS), which cause further oxidative stress to cancer cells [69]. A graphic illustration of the method is presented in Figure 3. The size of the NPs was about 130 nm, and their LSPR was at 800 nm, and they also demonstrated an efficient photothermal conversion ability.

The scientific team proceeded with in vitro tests on MCF-7 breast cancer cells and MCF-10A human breast epithelial cells to assess cytotoxicity, where both GNCs and GNCs@NO were implemented. The results showed a dose- and time-dependent cytotoxicity on both MCF-7 and MCF-10A cells. For concentrations <12.5 g/mL, cell viability remained >90%. Consequently, this concentration was chosen as the optimal dose for the following experiments. All combinations of NP, RT, and PTT therapies were then conducted, yielding some significant results.

First, the combination of NPs with PT + RT led to a significant rise in cell destruction for both GNCs and GNCs@ NO, compared to NPs + RT and NPs + PTT alone, proving the importance of the synergetic treatments. Additionally, an increase in ROS presence was apparent in both GNCs@NO + RT and GNS + RT, indicating a strong radiosensitization effect. The presence of RNS was noted only in the combinational therapy of GNCs@NO + RT + PTT, as both thermal-induced release of NO and RT-mediated ROS production are needed. Under these conditions, the cell viability was significantly reduced, and the colony formation measured with a clonogenic survival assay was very low. The cytotoxicity Sulforhodamine B (SRB) assay showed that GNCs@NO + RT + PTT provoked the highest tumor cell killing, causing an 80% reduction in cell viability, compared to the moderate effects caused by PTT and RT alone. The chain of events of combinational treatment is that the temperature rise led to the release of NO gas, while at the same time alleviated tumor hypoxia. When RT was additionally applied, the combinational therapy led to the lowest colony formation, ensuring cell growth inhibition.

The most prominent antitumor effect was demonstrated in the in vivo tests conducted on Triple Negative Breast Cancer (TNBC)-bearing rodents. NPs were administered by injection to the tail vein of MCF-7 xenografts and different combinations of NPs, PTT, and RT were once again tested. The results were as expected. While in the control group, tumor volumes grew rapidly, reaching up to 1200 mm^3^, in the GNCs@NO + NIR + RT group, tumors were almost undetectable by day 18. Most importantly, no systemic toxicity was detected from histopathology images of the livers, spleens, kidneys, and hearts of the mice, complying with the overall body weight rise of the xenografts.

Gold nanocages (GNCs) are another type of NP that enable theragnostic applications. In a study by Tang et al., PEG and CD44 antibodies were conjugated to GNCs (CD44-PEG-GNCs) for both biocompatibility and tumor-targeting enhancement [70]. CD44 receptors are overexpressed in 4T1 breast cancer cells; thus, antibody conjugation enables active tumor targeting. CD44-PEG-GNCs’ cellular uptake was significantly higher than PEGylated GNCs in 4T1 cells (~1.8×), because of the CD44 receptor-mediated endocytosis, and TEM images confirmed intracellular localization of the NPs, especially accumulated near organelle membranes. The cytotoxicity of naked GNCs was minimal, with cell viability remaining >80% over a 48 h period. Gold nanocages (GNCs) demonstrated excellent photothermal conversion efficiency (PCE), with temperatures reaching 57–58 °C within 16 min under 808 nm laser irradiation (2.5 W/cm^2^). Notably, CD44-PEG-GNCs exhibited slightly higher heating efficiency than PEG-GNCs, likely due to increased cellular retention. Cell viability was reduced to 38.6% with CD44-PEG-GNCs + PTT at 5 W/cm^2^, while CD44-PEG-GNCs + RT caused significant cell apoptosis and cell survival dropped to 67.8%.

The synergy of the two methods was tested only in vivo, on female BALB/c mice bearing 4T1 breast tumors. The xenografts were treated differently with all combinations of PEG-GNCs and CD44-PEG-GNCs, PTT and RT, and GNCs. Intravenous (i.v.) and intratumoral (i.t.) injections were compared, with i.t. injections showing higher tumor accumulation. Hematoxylin and eosin (H&E) staining confirmed the enhanced antitumor effects of gold nanocages (GNCs) combined with photothermal therapy (PTT) and radiotherapy (RT). The control (PBS) and PBS + NIR/X-ray groups showed intact tumor cells with minimal necrosis, while PBS + PTT + RT induced only mild cell shrinkage. In contrast, PEG-GNCs (i.v.) + PTT/RT led to increased necrosis, karyorrhexis, and inflammatory infiltration, indicating enhanced tumor destruction. CD44-PEG-GNCs (i.v.) + PTT/RT further amplified these effects, causing extensive extracellular matrix degradation. Notably, intratumoral (i.t.) GNCs + PTT/RT resulted in near-complete tumor necrosis, confirming the synergistic efficacy of GNCs in enhancing RT/PTT-mediated tumor ablation.

Gold nanorods coated with cancer cell membranes (GNR@Mem) were developed to enhance tumor targeting, immune evasion, and therapeutic efficacy in RT and PTT [71]. The plasma membrane coating from KB oral squamous cancer cells improved nanoparticle retention at tumor sites by enhancing homotypic tumor targeting, leading to 3.5× higher tumor accumulation compared to PEGylated GNRs (GNR@PEG). In in vitro studies, GNR@Mem + PTT (980 nm NIR II, 0.5 W/cm^2^) decreased cell viability to 5.8%, while GNR@Mem + RT (4 Gy) reduced viability to 22.6%. The combination of RT + PTT further lowered viability to 1.4%, with 55.3% of cells undergoing apoptosis, as confirmed by Annexin V-FITC/PI flow cytometry analysis. This highlights the strong therapeutic synergy of the combinational treatment.

In in vivo studies using BALB/c nude mice with KB xenograft tumors, RT alone provided moderate tumor suppression, while PTT alone induced partial tumor shrinkage. However, AuNR@Mem + RT + PTT resulted in 95.6% tumor inhibition, with complete regression in four of five mice over a 28-day period. On the contrary, tumors in the control group grew rapidly, reaching ~2300 mm^3^.

Histological analysis confirmed extensive necrosis and minimal damage to healthy organs, supporting the biocompatibility and selectivity of GNR@Mem. These findings highlight GNR@Mem as a potent theragnostic platform, integrating RT, PTT, and tumor-specific targeting for enhanced cancer treatment outcomes.

Table 1 summarizes the main characteristics of the studies conducted with GNPs discussed in this section. 

**Table 1 cancers-17-02295-t001:** Studies conducted with GNPs.

NP Type	NP Size (nm)	In Vivo/In Vitro	Concentration	Laser Conditions	X-Ray Conditions	Results	Citation
Alginate-coated gold NPs (Au@Alg-DA)	8.7 ± 1.3 nm (TEM), 95 nm (DLS)	In vitro (4T1 breast cancer cells) and in vivo (Balb/c mice)	500 µg/mL	808 nm, 1 W/cm^2^, 5 min (in vitro)/2 min (in vivo)	4 Gy, 6 MV	Triple therapy: ~35% viability; strong ROS and tumor suppression.	[67]
AuPt NPs functionalized with folic acid (AF)	16 nm	In vitro (4T1 breast cancer cells) and in vivo (BALB/c mice)	50–200 µg/mL	808 nm, 1 W/cm^2^, 5 min	6 Gy	Triple therapy: 19% viability; 94% tumor inhibition in vivo.	[72]
Gold nanocages (GNCs) loaded with NO donor (GNCs@NO)	110.4 nm (DLS)	In vitro (MCF-7 breast cancer cells) and in vivo (MCF-7 xenograft in nude mice)	12.5 µg/mL	808 nm, 1 W/cm^2^, 5 min	1 Gy/min, 5 min	RT + PTT + NO: 80% cell kill; full tumor regression.	[68]
Folate-conjugated gold NPs (F-AuNPs)	5–20 nm (TEM), 18 nm (DLS)	In vitro (KB mouth epidermal carcinoma cells)	20–40 µM	532 nm, 0.5 W/cm^2^, 15 min	6 MV, 2 Gy	RT + PTT: 54% viability; 27.8% apoptosis; DEF 1.46.	[63]
Fe_3_O_4_@Au/reduced graphene oxide (rGO) nanostructures	10–60 nm (TEM), avg. 29.4 nm	In vitro (KB oral squamous carcinoma cells)	5, 10, 20 µg/mL	808 nm, 1.8 W/cm^2^, 5 min	2 Gy and 4 Gy, 6 MV	RT + PTT: 11.9% viability; strong ROS and thermal effect.	[73]
Acid-induced aggregated gold NPs	~40 nm (individual), ~1000 nm (aggregates)	In vitro (MCF-7 breast cancer cells) and in vivo (MCF-7 xenograft in nude mice)	50 µg/mL	808 nm, 2.0 W/cm^2^, 10 min	4 Gy, 137 Cs, 662 keV	SER = 1.68; strong PTT; tumor regression, no recurrence.	[65]
Cancer cell membrane-coated gold nanorods (GNR@Mem)	68 × 11 nm	In vitro (KB oral squamous carcinoma cells) and in vivo (KB xenograft in nude mice)	5 mg/kg (in vivo)	980 nm, 0.5 W/cm^2^, 5 min	4 Gy	RT + PTT: 1.4% viability; 95.6% tumor inhibition.	[71]
CD44-targeted gold nanocages (CD44-PEG-GNCs)	58.14 ± 4 nm	In vitro (4T1 breast cancer cells) and in vivo (4T1 xenograft in BALB/c mice)	3 nM	808 nm, 2.5 W/cm^2^, 5 min	2, 4, 6, and 8 Gy, 6 MV	RT + PTT: near-total necrosis; high tumor targeting.	[70]
Iodine-labeled gold nanorods ([^131^I]Au NRs-PEG)	70.6 ± 8.4 × 10.8 ± 3.5 nm	In vitro (MCF-7 breast cancer cells) and in vivo (MCF-7 xenograft in nude mice)	50 μCi ^131^I, 80 μg Au per mouse	808 nm, 1 W/cm^2^, 5 min	4 Gy	RT + PTT: enhanced suppression; dual imaging use.	[74]

### 2.2. Pt NPs

In a similar manner to gold NPs, Pt NPs are also plasmonic metal NPs with a high atomic number (Z = 78), which makes them an ideal candidate for RT + PTT combined therapies. Moreover, Pt-based compounds are already in clinical use as radiosensitizers (e.g., cisplatin) [75].

In a study published in 2020, platinum NPs (PtNPs), with an average diameter of 12.2 ± 0.7, were implemented for the treatment of melanoma cancer cells in vitro (B16/F10 cell line) [76]. For the purposes of this study, a continuous-wave diode laser of 808 nm and 6 MV LINAC were used. The aim of the experiments was to monitor any enhancement in the treatment.

Throughout the experiments, concentrations of 10, 50, 100, and 250 μg/mL were used. Laser irradiation had a duration of 10 min per well, with a power density of 1.0 W/cm^2^ and 1.5 W/cm^2^. The delivered radiotherapy doses were 2, 4, and 6 Gy. These different conditions were tested either alone or combined under different setups, to evaluate the individual and synergistic effects of PtNPs, PTT, and RT. Cell viability was calculated with an MTT assay after 24 and 72 h.

PtNPs were tested alone to assess their biocompatibility, and it was determined that concentrations up to 250 μg/mL did not cause any cytotoxicity effects. Furthermore, the cytocompatibility of laser light was validated via variability assay after laser irradiation. Additionally, in the treatment group of PtNPs + PTT, laser sensitization was observed for both the power densities of 1.0 and 1.5 W/cm^2^; however, the viability in both cases was >50% for the maximum concentration of 250 μg/mL. The use of PtNPs led to a significant increase in the therapeutic outcome in the PtNPs + RT group, with a cell viability of 13.7% (250 μg/mL PtNPs + 6 Gy) after 72 h compared to 39% with RT alone. The RT enhancement was also validated with the use of a fluorescent probe called 2′,7′-dichlorodihydrofluorescein diacetate, a measure of ROS generation. The number of ROS was the highest in the combination group treated with 100 μg/mL PtNPs + 1.5 W/cm^2^ + 6 Gy, while simultaneously, there was an even greater decrease in cell viability, reaching 10%. It is interesting that, to reach a similar reduction in cell viability with NPs + RT, the NP concentration was 2.5× higher (250 μg/mL) compared to the 100 μg/mL used in the triple treatment (NPs + RT + PTT).

In the study described above, the NPs consisted plainly of Pt. However, in other studies, Pt was used in combination with Au, as they appear to have complementary physicochemical properties. While GNPs exhibit a strong surface plasmon resonance, platinum has great light absorption efficiency and thermal conductivity, enhancing the generation of heat and the stability of the structures more than gold alone. Additionally, the high biocompatibility of Au prevents the degradation of Pt in physiological environments.

Tang et al. developed gold–platinum alloy NPs (AuPt NPs), functionalized with folic acid (AF), to enhance tumor targeting and enable synergetic PTT and RT, with a size of 16 nm [72]. Initially, the NPs were tested under 808 nm laser irradiation to identify their concentration and power density dependence, the stability of their photothermal conversion performance, and their photothermal conversion efficiency (PCE). Compared to AuPt NPs of different sizes and other conventional photothermal agents, AF demonstrated a high and stable PCE of 46.84%. The enhanced results observed with AF compared to AuPt NPs can be attributed to an increased cellular uptake, which was captured by TEM, as seen in Figure 4.

Subsequently, the therapeutic potential of AF was assessed both in vitro, using the 4T1 murine breast cancer cell line, and in vivo, in 4T1 tumor-bearing BALB/c mice. In the in vitro tests, there was a significant increase in ROS production (77.81%) for AF + RT + NIR, and a significantly reduced cell viability, as well as increased expression of γ-H2AX, indicating a large amount of DNA DSBs. The in vivo studies demonstrated a high tumor inhibition rate of 94%. While in the control group the tumors grew from 207 mm^3^ to 1449 mm^3^, in the AF + RT + NIR group, the tumor size was reduced to 87.4 mm^3^. Furthermore, within 5 min of irradiation (808 nm, 1 W/cm^2^), the tumor temperature reached 50 °C. Histological analysis of tumors treated with the triple combination showed high ROS levels, reduced proliferation (measured with Ki67 staining), and extensive necrosis.

A combination of Au and Pt was also used in a study by Liu et al., where PEGylated Au@Pt nanodendrites acted as a multifunctional theragnostic agent for synergistic RT + PTT and CT imaging [77]. The enhanced NIR absorption offered by the dendritic Pt nanostructures led to a higher PCE compared to spherical structures. PEGylation enhanced biocompatibility and improved dispersity in biological environments. In vitro studies on 4T1 breast cancer cells combined Au@Pt NDs with RT (4 Gy), leading to a reduction of cell viability to 32%. The combination of PTT (808 nm laser, 1 W/cm^2^, 10 min) with Au@Pt NDs led to a 45% reduction. As expected, Au@Pt NDs + RT + PTT had the most important effect, decreasing cell viability to 30%, while live/dead staining confirmed an increased level of apoptosis in the treatment group. Au@Pt NDs were then tested as CT contrast agents, exhibiting strong contrast (28.6 HU/mL/mg), better than clinically used iodine-based contrast agents. These results strongly suggest the potential of the NPs as theragnostic agents and lay the foundation for future research.

The following table (Table 2) is a summary of the experiments described in this section.

### 2.3. Cu NPs

Another type of NP that has been implemented for synergistic RT + PTT treatment is Cu NPs. One characteristic of solid tumors, apart from their hypoxic environment, are their elevated levels of glutathione (GSH). Glutathione is an important antioxidant that prevents damage caused by ROS to the DNA and other important components of the cells [78,79]. Cu NPs are capable of GSH depletion, as in the acidic tumor environment, Cu^2^^+^ ions are released, which then consume intracellular GSH [80].

In a 2022 study, researchers developed a novel nanoparticle platform for combined RT and PTT by coating hollow CuS NPs with platelet cell membranes (PC) to target hypoxia and deplete intracellular glutathione (GSH) [81]. The core of the NPs consisted of hollow CuS NPs, which exhibited an increased photothermal efficiency in the NIR-I region (808 nm). The CuS NPs were coated with platelet cell membranes to ensure that the NPs would go undetected by the immune system and prolong the circulation time. Additionally, tumor targeting was ensured due to the affinity of platelets to inflammatory tumor microenvironments.

The NPs were tested in vitro on the undifferentiated colon carcinoma cell line CT26. Cell viability assays showed a reduction of 59% for the PC + NIR + RT group, compared to RT or PTT alone. In this group, ROS levels were elevated, while GSH levels were decreased. For in vivo validation, CT26 tumor-bearing mice were used. In the PC + NIR + RT group, tumor volume was reduced by over 80% compared to the control group. Regarding the temperature rise, the use of PC led to higher temperatures, reaching 51.3 °C, a temperature much higher than the one accomplished with non-biomimetic CuS NPs (~37 °C). In the case of PC, the number of NPs reaching the tumor site was much higher, and the higher concentration of localized NPs led to the greater temperature rise.

In another experimental study, a novel Cu-based nanosystem was developed, by embedding Cu-doped polypyrrole (CuP) NPs in an injectable hydrogel matrix (CH system) [82]. CH not only caused GSH depletion, thus enhancing ROS accumulation intracellularly, but also acted as a nanozyme, exhibiting enzyme-like catalytic activity, converting H_2_O_2_ to OH^−^. In the in vitro tests that followed, CH + RT + PTT led to a much-reduced cell viability of 10% compared to RT or PTT alone. Figure 5a demonstrates the extremely reduced colony formation ability of the cells in the CH + NIR + RT treatment group compared to the others. Furthermore, an increased ROS presence was documented under confocal laser scanning microscopy (CLSM) with a DCFH-DA fluorescence assay (Figure 5b). GSH levels were significantly decreased. In Figure 5c, DNA damage is assessed via γ-H2AX immunofluorescence staining across the different treatment combinations, with the most pronounced damage observed in the CH+NIR+RT group.

For the in vivo experiments, 4T1 murine breast cancer cells were subcutaneously injected into the right flank of female BALB/c mice. CH with a Cu concentration of 20 μg/mL was injected intratumorally for localized delivery. In the CH + RT + PTT group, there was a 90% tumor inhibition compared to controls. Tumor histological analysis with TUNEL staining and Ki-67 immunostaining demonstrated significant tumor cell apoptosis and decreased cell proliferation.

A single-compartment nanoplatform was developed for combinational internal RT and PTT against Anaplastic Thyroid Carcinoma (ATC) in an orthotopic mouse model [83]. The nanoplatform consisted of pegylated [^64^Cu]CuS NPs, combining the radiotherapeutic properties of ^64^Cu with the plasmonic properties of CuS NPs. For the purpose of this study, the following treatment groups were tested: no treatment, laser alone, PEG-CuS NPs alone, RT alone (PEG-[^64^Cu]CuS NPs), PTT alone, and combined RT + PTT. The results of RT/PTT led to a tumor growth inhibition of 83.14%, a percentage significantly enhanced compared to RT alone (74.96%) or PTT alone (50.87%). An increase in median survival was also noticed (44 days in RT/PTT compared to 36 days for RT alone and 35.5 days for PTT alone). In addition to its therapeutic function, ^64^Cu enables PET imaging, complementing the CT imaging typically associated with metal-based NPs. Thus, PEG-[^64^Cu]CuS NPs offered a theragnostic approach, by integrating PET/CT imaging into the therapeutic process. Images acquired with a Micro-PET/CT scanner showed that 50% of the NP dose was retained at the tumor site even after 48 h post-injection, enabling prolonged therapy. Furthermore, PET imaging enabled real-time monitoring of the NPs, which allowed for more accurate radiation dosimetry calculations and personalized treatment planning.

A summary of the conditions and results of the experiments that were conducted with Cu NPs is presented in Table 3.

### 2.4. Bismuth NPs

Bismuth NPs, having a high atomic number (Z = 83), are another feasible option for ROS generation and thus radio-enhancement [84]. Experimental studies conducted both in vivo and in vitro have shown a 6-fold increase in ROS production with X-rays, when Bi NPs coated with red blood cell membranes and folic acid were implemented [85]. In another study conducted by Jiao et al., Bi NPs coated with cellulose nanofibers were used in 4T1 breast cancer-bearing mice in combination with 10 Gy X-rays, leading to significant tumor suppression [86]. A mathematical model developed by Hossain et al. revealed a higher dose enhancement of Bi NPs compared to gold and platinum NPs [87].

Additionally, the strong NIR absorbance demonstrated by Bi NPs makes them suitable for RT and PTT synergetic treatment. Yu et al. used DSPE-PEG-coated NPs for enhanced CT imaging, and they were successfully tested for their photoacoustic imaging capability [88]. In the same study, significant tumor growth inhibition was observed in vivo when NIR PTT (808 nm laser, 1.0 W/cm^2^, 10 min) was combined with 4 Gy X-rays. Additionally, in a study conducted by Yang et al., Bi NPs with lipid coatings were used for multimodal therapy (CT imaging, PA imaging, PTT, and RT) in vivo, reaching high tumor ablation rates [89].

Bismuth Sulfide Nanorods (Bi_2_S_3_) were tested in vitro and in vivo [90]. Enhanced RT efficacy was noticed, with an increase in ROS generation and DNA damage in tumor cells confirmed via comet assay and γ-H2AX staining. The SER with RT was 1.22, and reached up to 1.34 when RT was combined with NIR laser irradiation (808 nm, 2 W/cm^2^). Furthermore, the combination led to an increase in DSBs, and the reduced activity of the PARP and Rad 51 DNA repair proteins.

Heat generation improved tumor oxygenation in vivo in 4T1 breast cancer-bearing mice, reducing the expression of HIF-1a proteins. Bi_2_S_3_ + RT + NIR had the most significant tumor growth inhibition among the different treatment groups, while Kaplan–Meier survival analysis showed a significantly prolonged survival. The same NPs were also tested using dual-modal imaging for future theragnostic applications. In PA imaging, intravenous injection of NPs led to a strong contrast enhancement effect, while in CT imaging, they had a higher HU value than clinically used contrast agents like iopromide. What is also of great importance is that NPs enhanced PTT and RT treatment-inhibited tumor invasion and metastasis via the downregulation of the VEGF and CD34 angiogenic factors. H&E staining showed minimal toxicity in the liver, kidney, and spleen, and blood markers for kidney and liver function remained normal.

In a similar manner, Zhang et al. developed the nanodrug named PIBD (PLGA@IR780-Bi-DTPA) by the combined loading of Bi-DTPA and IR780 on poly(lactic-co-glycolic) acid PLGA [91], which is a biosafe material approved by the Food and Drug Administration (FDA) [92]. IR780 is a lipophilic cation that selectively accumulates in the mitochondria of cancer cells due to their higher transmembrane potential. It is also used for PTT tumor ablation by converting NIR light (808 nm) into heat, while it can be used for contrast enhancement in photoacoustic imaging (PAI) and near-infrared fluorescence imaging (NIRF). A graphical abstract of the method is presented in Figure 6.

In vitro tests conducted on the 4T1 mouse breast cancer cell line demonstrated a significant reduction in cell viability (3.24%) in the PIBD + RT + NIR group, while RT alone, PIBD alone, and PTT alone had significantly weaker effects. Apoptosis analysis conducted with flow cytometry showed an apoptosis of 94.26%. In vivo tests conducted on 4T1 breast cancer tumor models showed nearly complete tumor elimination, as the tumor volume was undetectable in the PIBD + RT + NIR group. PIBD + RT and PIBD + PTT significantly reduced tumor size but did not eliminate tumors. Photoacoustic imaging was used to detect tumor blood oxygenation, which rose to 8.05% from 2.96%, indicating improved hypoxia alleviation. With histological analysis (H&E, TUNEL, and PCNA staining), extensive tumor apoptosis transpired, as almost no remaining viable cells were detected. No significant major organ toxicity was observed, and bismuth clearance was complete after 72 h. However, the Combination Index was CI = 0.662, proving a moderate synergy among RT and PTT.

In a recent study by Ma et al., solid bismuth (Bi) nanospheres were synthesized through the reduction of Bi^3+^ ions and subsequently etched with dimethyl sulfoxide (DMSO) to produce porous Bi (pBi) nanospheres modified with polyvinylpyrrolidone (PVP) [93]. The PVP coating enhances their water solubility and biocompatibility while also enabling the porous structure to serve as an efficient carrier for anticancer drugs. The local temperature rise caused by PTT facilitated the pH- and heat-responsive release of the chemo-drug solely in the tumor environment. In this study, pBi nanospheres were tested as multifunctional agents for RT + PTT enhancement and Doxorubicin (DOX) chemo-drug delivery carriers. Additionally, they were tested as CT imaging contrast agents for tumor localization. In vitro tests confirmed a sufficient radiosensitization mechanism and a high photothermal conversion efficiency (48.5%). Additionally, in vivo tests on HeLa tumor-bearing mice resulted in complete tumor growth inhibition with pBi/DOX + RT + PTT, with no significant systemic toxicity. The CT signal intensity was strongly correlated with Bi concentration, offering high-resolution tumor imaging suitable for treatment guidance.

Table 4 presents a summary of the studies described above.

### 2.5. Magnetic NPs

Magnetic NPs can be used for magnetic hyperthermia, as many studies have implemented them with the use of alternating magnetic fields to induce temperature rise [94]. In this section, however, the magnetic properties of the NPs described are solely used for magnetic resonance imaging (MRI) purposes. In the present work, heat generation is studied exclusively via the mechanisms of PTT.

A research team tried to advance the combination of RT + PT by integrating MRI agents for tumor detection and monitoring [95]. The nanoplatform developed consisted of graphene oxide (GO), which is the photothermal agent for heat generation, gold NPs for enhanced radiosensitization, and superparamagnetic iron oxide (SPIO) as an MR-compatible imaging agent. The multifunctional theragnostic graphene oxide nano-flakes (GO-SPIO-Au NFs) were tested both in vivo and in vitro to assess their therapeutic and imaging capabilities. The in vitro tests were conducted to evaluate the cytotoxicity of the NFs on CT26 colon cancer cells, as well as to assess the radiosensitization and photothermal effects. The MTT assay used to evaluate cell viability demonstrated the minimal cytotoxicity of GO-SPIO-Au NFs alone (90% viability). Furthermore, the synergistic effect of NFs + RT + PT showed a much-reduced cell viability of 18%, compared to PTT and RT alone (50% and 45%, respectively). Furthermore, NFs + RT + PT led to a significant increase of ROS (33%), and an important reduction in clonogenic survival.

For the in vivo tests, female BALB/c mice were injected subcutaneously into the right flank with CT26 cells. The tumors were treated when their size reached 100 mm^3^. The route of administration of GO-SPIO-Au NFs was intravenous injection. The group that received combination therapy showed significant tumor growth inhibition, as the tumors were almost eliminated with no recurrence in the following 60-day period. Tumor imaging was performed for pre-treatment and for post-treatment monitoring. A T2-weighted MRI scan was conducted, as the contrast induced by the SPIO NPs in the nano-flakes shortened the T2 relaxation time and enhanced tumor visibility. Overall, apart from the significant synergistic effect of NF + RT + PTT both in vitro and in vivo, and the lack of toxicity, this study highlighted the systems’ theragnostic potential.

Magnetic-guided nanocarriers, consisting of iron–cobalt NPs with polydopamine and IR-780 (INS NPs), were used for the synergistic treatment of TNBC [96]. In this experimental study, magnetic NPs were used not for imaging, but for targeted delivery with the use of external magnetic fields. Firstly, in vitro studies were conducted in 4T1 mouse mammary carcinoma cells and normal cell lines to assess cytotoxicity, cellular uptake, and the synergistic therapeutic effects of INS NPs. Cytotoxicity was measured by performing a CCK-8 assay, and showed that while the toxicity of INS NPs in normal cells was minimal, it was quite significant in 4T1 cells, especially under combined treatment. Additionally, clonogenic survival assays demonstrated the most significant colony formation reduction in the INS NPs + RT + PTT group, compared to the already reduced survival observed in the INS NPs + RT and the INS NPs + PTT groups. The highest intracellular levels of ROS were once again present in the INS NPs + RT + PTT groups. Furthermore, cellular uptake was measured with fluorescent imaging and the internalization of the NPs was proven to be very efficient in the cancer cells.

The in vivo experiments that followed aimed to evaluate the therapeutic effect and the safety of the INS NPs’ combination therapy, but also aimed to evaluate the distribution of the NPs and the tumor-targeting efficacy of their method. INs NPs consist of a superparamagnetic FeCo core, which makes the NPs highly responsive to an applied external magnetic field but lose their magnetism when the field is no longer applied. During the experimental process, 10 mg/kg of NPs were administered intravenously via the lateral tail vain of female BALB/c mice that were formerly implanted with 4T1 tumor cells subcutaneously and had developed a tumor of 50 mm^3^. The external magnetic field was applied with a neodymium magnet placed externally to the tumor post-injection. The magnet remained in place for 1–2 h, to ensure that there was enough time for NP retention at the tumor site. Fluorescence imaging, conducted with the fluorescence signal from IR-780 dye conjugated to the NPs, showed increased NP accumulation and retention to the tumor site for up to 168 h. Compared to the non-magnetic group, the concentration of INS NPs was two to three times higher inside the tumor. Inferentially, the increased concentration of NPs inside the tumor can reduce off-target effects and systemic toxicity, leading to a well-targeted treatment. It is worth mentioning that 60% of tumors were eradicated in the in vivo studies. Nevertheless, the prolonged retention that was demonstrated in this study implies that the NPs could be utilized over multiple treatment cycles with just a single administration. Throughout the treatment, no significant weight loss or organ toxicity was observed, confirming the high biocompatibility of INS NPs.

In this section, two distinct methods of imaging were integrated in the RT + PTT treatment process: MRI and fluorescent imaging. A third study combined both by developing multifunctional NPs for bioimaging-guided and photothermal-enhanced radiotherapy (RT) of non-small cell lung cancer (NSCLC) [97]. The core of the NPs consisted of polymethacrylic acid (PMAA) with Fe(III) and Cypate, an NIR fluorescent dye. Their unique coating is a mesenchymal stem cell membrane, which does not only offer immune evasion and thus longer circulation times, but also facilitates active tumor targeting via chemokine interactions [98]. Cyp-PMAA-Fe@MSC NPs were tested in vitro on LLC1 lung cancer cells. The NPs demonstrated low toxicity in normal cells, while in cancer cells, cell viability was significantly reduced when combinational treatment was applied. Cellular uptake was measured with flow cytometry and confocal microscopy. The uptake of MSC-coated NPs was more efficient when compared to RBC-coated NPs. Furthermore, the radiosensitization effect of the NPs was proven, as the γ-H2AX assay conducted on the RT + Cyp-PMAA-Fe@MSCs showed an increase in the number of DSBs. Additionally, the triple combination group exhibited the highest ROS levels and the lowest clonogenic survival rate.

The in vitro tests were followed by the in vivo application of the combinational treatment (Figure 7). The experiments were conducted on LLC1 tumor-bearing BALB/c mice that were injected intravenously with 150 μL of a concentration of 2 mg/mL PMAA. The rodents received PTT with an 808 nm laser and, consequently, 8 Gy of RT. The tumor volume increased by 2.2-fold in the PTT group and 3.3-fold in the RT group over 16 days, whereas the RT + PTT + Cyp-PMAA-Fe@MSCs group exhibited a 37% reduction in tumor size compared to the initial volume. Fluorescence imaging conducted on the rodents showed that the signal of the tumor sites in the Cyp-PMAA-Fe@MSCs group was 21% higher than in the RBC-coated group, proving the effectiveness of the NPs in tumor targeting. High tumor accumulation was also confirmed by MRI scans. Low-molecular weight Fe(III) complexes are widely used as MRI contrast agents [99]. Therefore, the NPs developed in this research work were also used for contrast-enhanced imaging of the tumors. The T1-weighted MRI signal was decreased by 30% post-injection, implicating, once again, intratumoral NP localization.

In another study, Yong et al. introduced ultrasmall (~3.5 nm) gadolinium polytungstate nanoclusters (GdW@BSA NCs) as multifunctional theragnostic agents [100]. The NPs were embedded with a Bovine Serum Albumin (BSA) coating to improve biocompatibility, to ensure uniform dispersion in biological fluids by preventing aggregation and help in rapid elimination from the body by renal clearance. GdW@BSA nanoclusters are suitable as radiosensitizers, as W and Gd are both high-Z materials, they have strong NIR absorption, and they can also be implemented for dual-imaging as they can work as both CT and MRI contrast agents. In vitro tests were conducted on 16HBE cells (Human Bronchial Epithelial Cells) as the healthy, non-cancerous control and HeLa cervical cancer cells. The 16HBE cells maintained high viability (>90%) even at high nanoparticle concentrations (1 mg/mL), confirming biocompatibility of the nanoclusters in normal cells. Additionally, the 16HBE cells internalized fewer NPs than HeLa cancer cells, indicating preferential accumulation in tumors, suggesting low off-target toxicity. When the GdW@BSA nanoclusters were used in combination with PTT and RT, there was a nearly complete cancer cell destruction, with only 5% survival, proving a strong PCE (~55 °C in 10 min at 1 W/cm^2^) and radiosensitization capability. It was deduced that the effect of the combinational treatment was even stronger than the already significant effect of the GdW@BSA + PTT combination (10% survival). GdW@BSA + RT led to 60% survival. The strong effect observed in the GdW@BSA + PTT combination can be considered to be because of the significantly strong photothermal conversion efficiency of the NPs.

In vivo tests on BEL-7402 tumor-bearing BALB/c nude mice (human liver cancer model) did not include a GdW@BSA + PTT + RT group; however, there was a very effective tumor suppression in the GdW@BSA + RT group, with complete tumor elimination and no recurrence for 14 days. RT alone reduced tumor volume but could not eliminate tumors. Gadolinium polytungstate nanoclusters (GdW@BSA NCs) demonstrated strong T1-weighted MRI contrast enhancement due to their high relaxivity (r = 9.45 s^−1^mM^−1^), significantly outperforming commercial Gd-DTPA (3.90 s^−1^mM^−1^). In MDA-MB-231 tumor-bearing mice, intravenous injection of GdW@BSA NCs resulted in sustained tumor signal enhancement, with an MR signal two times higher than Gd-DTPA after 2 h. This indicates effective passive tumor targeting and a longer retention time, making them superior to conventional contrast agents. Additionally, significant renal clearance of the nanoclusters was observed, suggesting minimal long-term toxicity risks. Beyond MRI, GdW@BSA NCs also functioned as efficient CT contrast agents due to their high-Z elements (Gd and W). Their X-ray absorption coefficient (19.2 HU mM^−1^) was three times higher than commercial iopromide (6.09 HU mM^−1^), providing superior CT imaging contrast. In vivo CT imaging of MDA-MB-231 tumors confirmed strong tumor signal enhancement after intratumoral injection. These findings highlight GdW@BSA NCs as dual-modal contrast agents, enabling precise tumor localization and real-time imaging for theragnostic applications in cancer treatment, while their ultrasmall size enabled rapid renal clearance, reducing long-term toxicity concerns.

Table 5 presents a simple summary of the experiments described in this section.

### 2.6. Other Metallic NPs

In a study by Shakerian et al., the synergistic effects of RT and PTT were examined in vitro, using Fe_3_O_4_@Au/reduced graphene oxide (rGO) nanostructures on KB oral squamous carcinoma cells [73]. The NP complex was synthesized with a hydrothermal reaction method and characterization procedures confirmed spherical Fe_3_O_4_@Au NPs with diameters (10–60 nm) well-dispersed on rGO nanosheets. The scientific team synthesized two different formulations of the NPs, with 10 wt% and 40 wt% rGO. The PCE of the 40 wt% rGO formulation calculated at 61% was superior to the other, indicating them to be good candidates for PTT. Furthermore, in both the PTT alone and RT alone experiments, a higher graphene content improved RT efficiency and enhanced NIR absorption and consequently, heat generation. Among the tested concentrations, 20 μg/mL of Fe_3_O_4_@Au/rGO (40 wt% rGO) was the most effective in the combined RT + PTT treatment, reducing cell viability to 11.9%. In comparison, RT alone resulted in 50.2% viability, and PTT alone reduced it to 27%.

Jiang et al. conducted a study to evaluate the potential of zirconium carbide NPs (ZrC NPs) in combined PTT and RT for the treatment of triple-negative breast cancer (TNBC), in vitro and in vivo [101]. ZrC NPs were firstly functionalized with bovine serum and folic acid (FA) to increase their biocompatibility and tumor targeting. The treatment modalities used were a near-infrared (NIR) laser (808 nm) for PTT and a 6 MV LINAC for RT. ZrC NPs exhibited a very high PCE of 40.51%.

The in vitro study was conducted with 4T1 breast cancer cells and the treatment groups were control (no treatment), PTT, PTT + ZrC NPs, RT, RT + ZrC NPs, and ZrC NPs + PTT + RT. An 808 nm NIR laser (1.5 W/cm^2^, 10 min) was used for PTT, while 4 Gy X-ray irradiation (6 MV LINAC) was used for RT. Cell viability was calculated for every treatment group via MTT assay, with the highest being 109% for PTT alone, meaning that there was a small cell growth most likely due to the mild temperature rise. In the ZrC + RT group there was a very strong radiosensitization effect, with SER = 1.8. For PTT + ZrC NPs and RT + ZrC NPs, the cell viability was 39% and 38.7%, respectively. ZrC NPs + PTT + RT had the strongest cancer-killing effect, with only 5% cell viability. Furthermore, DNA damage was assessed with γ-H2AX fluorescence, which is a measure of double-strand breaks, showing significant DNA damage with ZrC NPs + PTT + RT compared to the other groups.

In the in vivo study, the research team used a mouse model with a subcutaneous 4T1 tumor. The treatment groups were the same as in the in vitro experiments. The tumors were treated when they reached the size of 200 mm^3^. NPs were administered intravenously (0.5 mg/mL, 100 μL). The mice were then monitored for a 14-day follow-up period. Tumor volume measurements showed a 4-fold increase of the initial tumor volume in the control group, moderate tumor suppression in the ZrC NP + PTT group, but with high local recurrence, and significant tumor suppression with ZrC NPs + RT. However, the highest tumor growth inhibition and suppression was a result of ZrC NPs + RT + PTT. Finally, the researchers noted a reduced growth rate of distant tumors in treated mice (untreated tumors developed on the other side of the treated tumors), as well as an increase in the number of CD8+ tumor-infiltrating cancer cells and CD4+ helper T cells, which could indicate an immune activation related to the results of PTT.

In a study by Liu et al., RGD-PEG-PAA-MN@LM NPs were evaluated both in vitro and in vivo for the combined effects of photothermal therapy (PTT), photodynamic therapy (PDT), and radiotherapy (RT) in the treatment of hepatocellular carcinoma (HepG2) [102]. For the purposes of this study, three different types of NPs were tested. Firstly, liquid metal NPs (LM) of eutectic gallium–indium (EGaIn) were used for the baseline comparison as a PTT agent with no specific targeting. The second type of NP was RGD-PEG-PAA-MN, which contains polyethylene glycol (PEG) and a polyacrylic acid (PAA) copolymer, and is conjugated with an RGD peptide for enhanced tumor targeting (v3 integrins) and Metronidazole (MN), a hypoxia-responsive radiosensitizer. Finally, RGD-PEG-PAA-MN@LM combines both RGD-PEG-PAA-MN with an LM core and comprises the full nanoparticle system. The results showed that RGD-PEG-PAA-MN@LM brings together the advantages of both nanosystems, offering high tumor cell targeting, enhanced PTT due to LM’s efficient heat conversion ability, and enhanced radiosensitivity even in hypoxia because of the MN, as well as increased production of ROS when irradiated with NIR and X-rays (laser 808 nm, 2 W/cm^2^ for 6 min, 6 Gy X-rays).

In vitro experiments exhibited cell viability up to nearly 0% under hypoxic conditions when a concentration of 800 μg/mL was used. For the in vivo experiments, HepG2 tumor-bearing mice were injected with a concentration of 15 mg/kg. The key results of the study were a tumor volume reduction to 12.6 mm^3^ in the RGD-PEG-PAA-MN@LM group, compared to the control group in which the tumor volume was 790 mm^3^. Another significant finding was that, within 5 min of NIR irradiation, the tumor temperature was raised to 53 °C, causing thermal ablation. Once again, no damage to major organs was observed when they underwent histological tests.

In a study conducted by Yong et al., ultrasmall (3 nm) tungsten sulfide (WS_2_) quantum dots (QDs) were developed, following a green synthesis process [103]. The NPs demonstrated a high photothermal conversion efficiency (44.3%), enabling significant tumor heating (47 °C) and thermal ablation when irradiated with an 808 nm laser (1 W/cm^2^ for 10 min). In vitro tests of combinational treatments were conducted on BEL-7402 human liver cancer cells. WS QDs + RT + PTT led to only 6% cell survival, compared to 31% and 39% for WS QDs+ RT and WS QDs + PTT, respectively. RT monotherapy, without any NPs implemented, reduced cell viability to 75%, a very high number compared to the results of combinational therapies.

These promising in vitro results were successfully reproduced in vivo, where the combined RT + PTT approach resulted in complete tumor regression with no recurrence for 22 days on BEL-7402 liver cancer tumor-bearing BALB/c nude mice. Beyond their therapeutic effects, WS QDs were tested as multimodal imaging agents. CT-W has high atomic number-enabled precise tumor localization, suitable for image-guided RT, and PA confirmed that NPs were passively accumulated to the tumor site, with the signal intensity doubling within 2 h post-injection. Cytotoxicity studies demonstrated that the QDs are well-tolerated, causing minimal toxicity to normal cells even at high concentrations, while in vivo blood tests showed normal liver and kidney function.

A summary of the above can be found in Table 6.

## 3. Discussion and Future Perspectives

The incorporation of nanotechnology in medical oncology is a rapidly growing field, offering new valuable solutions to the challenging battle against cancer. Additionally, NPs have renewed scientific interest for PTT, as metallic NPs can be used as efficient photothermal energy converters due to the unique LSPR effect. The two treatment modalities seem to have complementary actions, which can be harnessed with the use of double-purpose NPs. Although this review focuses primarily on the mechanistic and material aspects of NP-mediated RT–PTT, it is important to acknowledge that the therapeutic effectiveness of such approaches can vary depending on the cancer type. For instance, superficial and well-vascularized tumors such as breast, cervical, and head and neck cancers are particularly amenable to PTT due to easier laser penetration and nanoparticle delivery. In contrast, deep-seated or hypoxic tumors like pancreatic or brain tumors may require alternative delivery strategies or hybrid modalities. Several studies included in this review address breast, colorectal, and melanoma models, which reflect the current focus of preclinical RT–PTT research. Future efforts should include broader tumor type comparisons and personalized treatment adaptation based on tumor accessibility, radiosensitivity, and microenvironment characteristics.

A wide range of metallic nanoparticles have been explored for their potential in dual-function radiotherapy (RT) and photothermal therapy (PTT) applications. This review has particularly emphasized high atomic number (Z) metals such as gold (Au, Z = 79), platinum (Pt, Z = 78), bismuth (Bi, Z = 83), and tungsten (W, Z = 74), whose radiosensitization capabilities are closely correlated with their atomic number. Among them, gold nanoparticles (GNPs) have emerged as the most prominent, exhibiting excellent radiosensitizing efficiency, which allows for effective tumor control at reduced X-ray doses. Notably, gold-based NP platforms have progressed to clinical evaluation, bringing nanoparticle-enhanced RT closer to integration into standard oncologic care [104].

In addition to radiosensitization, these high-Z metals also contribute distinct photothermal properties. Au exhibits strong localized surface plasmon resonance (LSPR), enhancing its photothermal efficiency. Tungsten-based nanoparticles, such as WS_2_ nanosheets and quantum dots, have consistently demonstrated high photothermal conversion efficiency (PCE) across various formulations. Bismuth-based NPs also possess strong NIR absorbance, making them suitable candidates for the combined PTT and RT strategy discussed in this review. Although platinum nanoparticles alone have relatively modest photothermal properties, this limitation can be addressed through their incorporation into bimetallic structures (e.g., Au@Pt), which combine the photothermal performance of gold with the radiosensitizing benefits of platinum.

However, there are also metals like Cu (Z = 29) and Zr (Z = 40), in the form of ZrC, whose low atomic number would not rationalize their use as radiosensitizers. In the case of Cu, atomic number-related radio-enhancement is not as prominent, but its strong photothermal effect combined with the ROS production boosted by GSH depletion leads to an overall adequate synergetic effect. In a similar manner, the high PCE of ZrC leads to an efficient generation of heat, improving its radiosensitizing effects due to thermal damage, improved oxygenation, and ROS production. Its overall strong radiosensitization is depicted by an SER of 1.8, comparable to that of gold-based nanosystems.

In the present work, we have seen various functionalization techniques. Folate receptors are a biomarker for tumor cells, and they are overexpressed in many tumors. The conjunction of folate to GNPs leads to increased tumor uptake and enhanced apoptosis [105]. In a similar manner, CD44 receptors were targeted with CD44 antibodies on CD-PEG-GNC, and integrin receptors were targeted with RGD peptides on RGD-PEG-PAA-MN. In those cases, we can also see the importance of PEGylation, which promotes biocompatibility and thus ensures prolonged circulation, immune evasion, and reduced systemic toxicity [106]. An additional targeting strategy described in this review is the so-called homotypic targeting, by functionalizing GNRs with cancer cell membranes, and specifically, coating with MCF-7 cell membranes. The mechanism behind this method is that, by coating NPs with a membrane of the same type of cancer, they can recognize and bind to the cancer cells as they retain all the surface proteins and adhesion molecules used among cancer cell interactions [107]. In a study described in the current work, GNR@Mem NPs had a 3.5× tumor accumulation compared to PEG-GNRs. Another similar method was described with platelet membrane coating of CuS NPs, which increased biocompatibility but also induced inflammation targeting [108]. Those biomimetic methods allow for lower immunogenicity and reduced clearance, while they also allow for further functionalization. Some studies have also explored magnetic-guided nanocarriers, such as iron–cobalt (FeCo) NPs, which allow for precise tumor targeting using an external magnetic field, improving treatment accuracy and reducing off-target effects.

In the rapidly evolving field of nano-oncology, image-guided treatment techniques have become a cornerstone of clinical practice, with pre-treatment image-guided radiotherapy (IGRT) long established as a standard approach. Current research and clinical interest are increasingly shifting toward real-time IGRT, which offers dynamic tumor tracking and improved adaptive treatment precision. The integration of real-time IGRT, particularly with advanced modalities like MRI-guided systems, is expected to enhance the treatment accuracy and monitoring of NP biodistribution in vivo. In the present work, we have seen nanoplatforms that serve as multi-purpose agents, enabling real-time monitoring. In a clinical setting, IGRT can benefit significantly by NPs’ contrast enhancement, while offering the radiosensitizing and thermal effects that we have thoroughly mentioned previously.

In a study by Sinha et al., the rationale of hypofractionated radiotherapy—the delivery of larger doses per fraction over fewer sessions—in combination with hyperthermia is being discussed [109]. Hypofractionation is more convenient for patients, and it can be more biologically effective in certain tumors (especially when combined with radiosensitizers like NPs). Studies suggest that hyperthermia can sensitize tumors to high-dose radiation, and NPs can enhance both modalities synergistically. As PPT is a form of hyperthermia, hypofractionation can be a natural match for NP-based dual therapy.

So far, the studies described combine PPT with X-ray radiation. However, there are also alternative radiotherapy methods, like boron neutron capture therapy (BNCT). BNCT requires a boron-containing drug that selectively accumulates in tumor cells and low-energy neutron irradiation, which is usually applied via an external neutron beam (e.g., from a reactor or accelerator). The main limitation of BNCT is the localized delivery of boron. In a recent study by Ma et al., a molecular probe called IR780-MPBA was designed, enabling combined BNCT and PTT—all in one system [110].

Further future advancements will likely involve the integration of immunotherapy in RT + PTT synergetic treatment, as it offers a promising avenue for both local and systemic tumor control. As discussed earlier, RT and PTT can also trigger immunogenic cell death (ICD), releasing tumor-associated antigens and danger signals (DAMPs) that activate the immune system. Localized ROS production and heat-mediated membrane disruption can be amplified by the implementation of NPs, thus promoting antigen presentation. A well-accepted study has demonstrated that combining PTT with immune checkpoint inhibitors (e.g., anti-PD-1/PD-L1 or anti-CTLA-4) can transform non-immunogenic tumors into responsive tumors [111]. Additionally, nanoparticle platforms can be engineered to co-deliver immunoadjuvants or modulate immune checkpoints, offering a highly customizable strategy to prime antitumor immunity. These combinatorial approaches may be particularly valuable in metastatic or recurrent cancers, where durable systemic responses are needed.

## 4. Limitations

Despite its promising potential, NP-mediated RT–PTT therapy faces several limitations that must be addressed before clinical translation. First, achieving efficient and uniform nanoparticle accumulation within tumors remains a challenge due to biological barriers, including heterogeneous vascularization and enhanced permeability and retention effect variability [112]. Second, laser penetration depth is limited, restricting PTT efficacy in deep-seated tumors unless invasive fiber-optic delivery is used [113]. Additionally, there is a risk of off-target heating and potential damage to adjacent healthy tissues, particularly in anatomically sensitive regions. The long-term biodistribution, metabolism, and clearance of certain NPs (e.g., heavy metals like bismuth or copper) also raise safety concerns, particularly for repeat administrations [114,115]. Table 7 presents an overview of the advantages and disadvantages of the different metallic NPs that are described in the present work.

One important point to consider regarding the limitations of the current situation for PTT + RT + NP treatment is preclinical stage dominance. All the studies described above are conducted either in vitro, or in vivo on small animal models. There are not yet clinical trials conducted on human patients; therefore, there is a translational gap remaining between the experimental results and the possible efficacy of the treatment methods in a clinical setting. There is a lack of large-scale clinical data validating the safety and efficacy of RT–PTT combinations in humans, highlighting the need for rigorous preclinical-to-clinical translation strategies.

Many studies also rely on the EPR effect, which is more prominent in animal models than in human tumors; therefore, it remains uncertain whether equally adequate NP distributions could be achieved via the same ways of administration in clinical patients [56].

Another issue that arises from the present work is the lack of standardized protocols. Even though most studies have demonstrated significant results, there are many variations among the experimental conditions, like the laser parameters and radiation dose. Therefore, direct comparison among techniques is infeasible. In most studies, 808 nm near-infrared (NIR) lasers are used due to their deep tissue penetration. However, exposure times and power densities vary from 2–10 min and from 0.5 to 2.5 W/cm^2^, respectively. Alternative wavelengths that have been explored for improved penetration depth and heating efficiency are 980 nm and 1064 nm.

As emphasized earlier, the mechanism for the most efficient heating lies with the LSPR effect. For this to be achieved, the wavelength of the laser must be coherent with the peak resonance frequency of the NP’s surface oscillations. In most studies described in the present work, the scientific teams do not seem to take this into consideration when choosing their laser and NP combination. Additionally, radiation doses varied widely, with sources ranging from clinical 6 MV LINAC systems to low-energy gamma sources (e.g., ^64^Cu and 137Cs), making it necessary to standardize protocols for optimized treatment outcomes.

The standardization of protocols is also vital for clinical translation. Differences in radiation doses, laser intensities, and NPs create inconsistencies in outcomes. It is vital to establish dose–response studies and clinical guidelines to achieve regulatory approval. Regulatory approval is further complicated by the complex design of multifunctional nanoplatforms, which combine diagnostic, therapeutic, and targeting elements. Additionally, one major concern for clinical implementation is toxicity and clearance, an issue more prominent in heavy-metal-based NPs, which may accumulate in organs if not efficiently excreted [126,127]. Some strategies that are being thoroughly explored are biodegradable polymer coatings, renal-clearable ultrasmall NPs, and biocompatible hybrid nanostructures [128,129,130].

Another important limitation is the lack of long-term data. In animal models, it is known that rodents are euthanized when reaching specific ethical endpoint criteria [131], while in cases of tumor resection, they will most likely be euthanized for histological analysis to be conducted. Therefore, most of the studies monitor short-term tumor suppression. There are no reports about recurrence rates, long-term efficacy, and potential chronic toxicity.

## 5. Conclusions

In conclusion, the combinational treatment of RT and PTT with the use of metallic NPs has shown some very promising results in preclinical settings. The strong synergy among the two methods based on their complementary action has led to significant antitumor effects in vitro and in vivo. Nevertheless, further research aiming for the standardization of protocols and elimination of long-term safety concerns is crucial for clinical translation. As discussed in the present work, there are many challenges remaining before the implementation of RT + PPT treatment in standard clinical treatment. However, there are many future advancements that are likely to be integrated, as the most valuable aspect of nanomedicine is the ability to create versatile, multimodal nano systems.

In recent years, nanoparticle-mediated approaches have advanced the frontiers of both PTT and RT, and their synergistic integration represents a novel and promising direction in cancer treatment. The novelty of this review lies in its focused analysis of recent developments in multifunctional nanoplatforms that serve dual roles: enhancing radiation response while enabling localized photothermal ablation, often with concurrent diagnostic imaging. Unlike previous reviews that examine RT and PTT separately, this work consolidates recent advances specifically in RT–PTT combinatorial nanotherapeutics, highlighting their shared physical mechanisms, challenges, and translational opportunities.

Looking ahead, significant efforts are needed to optimize nanoparticle design for clinical use, including biocompatibility, tumor accumulation, and clearance. Future research should emphasize clinically translatable platforms with scalable synthesis, integrated imaging, and surface modifications for tumor targeting. The combination with immune modulation, such as checkpoint inhibitors or antigen release via heat and radiation, represents a particularly promising direction. Moreover, clinical trial frameworks are needed to rigorously evaluate the efficacy and safety of RT–PTT therapies in comparison with conventional treatment paradigms.

Overall, NP-mediated RT–PTT offers a highly controllable, tumor-specific, and potentially immunogenic cancer treatment strategy. Its successful translation into clinical practice will depend on addressing technical challenges, improving delivery efficiency, and building interdisciplinary collaborations across materials science, oncology, and imaging fields.

## Figures and Tables

**Figure 1 cancers-17-02295-f001:**
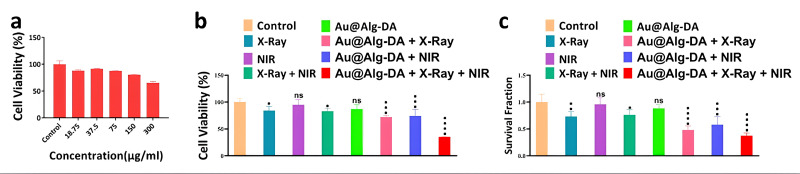
Assays conducted in vitro with Au@Alg-DA NPs: (**a**) cell viability (%) of HUVEC cells with varying concentrations of NPs, (**b**) cell viability (%) of 4T1 cells with and without irradiation, and (**c**) survival fraction of 4T1 cells with and without irradiation. Reprinted/adapted with permission from [67], 7 May 2025, distributed under a Creative Commons Attribution 4.0 International License.

**Figure 2 cancers-17-02295-f002:**
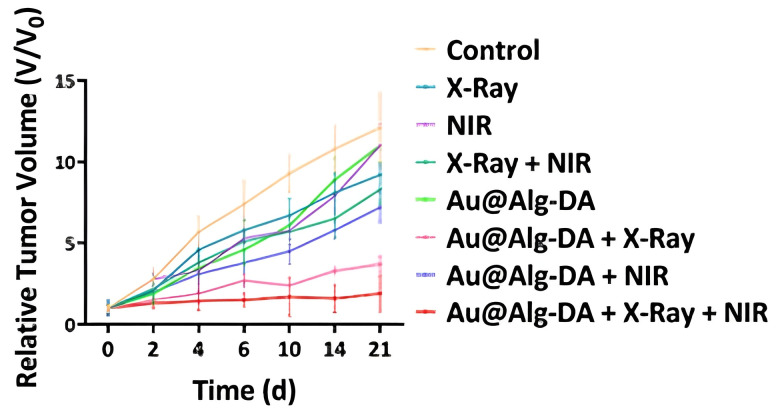
Tumor growth in different treatment groups as a measure of in vivo anticancer ability. Reprinted/adapted with permission from [67], 7 May 2025, distributed under a Creative Commons Attribution 4.0 International License.

**Figure 3 cancers-17-02295-f003:**
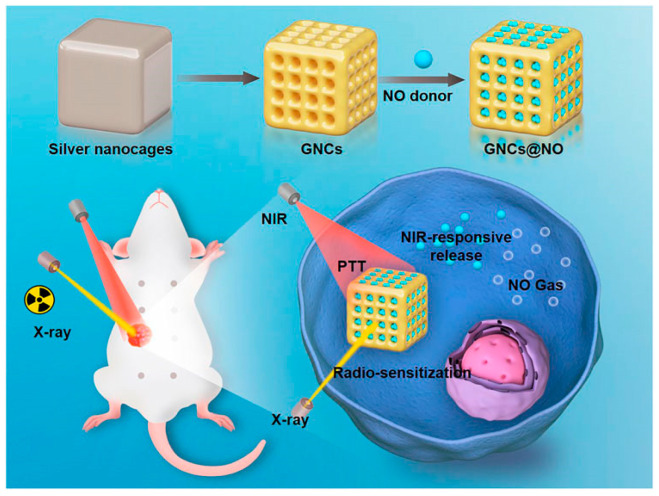
Fabrication of gold nanocages with nitric oxide (NO) donors for enhanced RT + PTT treatment. Reprinted/adapted with permission from [68], 7 May 2025, distributed under a Creative Commons Attribution 4.0 International License.

**Figure 4 cancers-17-02295-f004:**
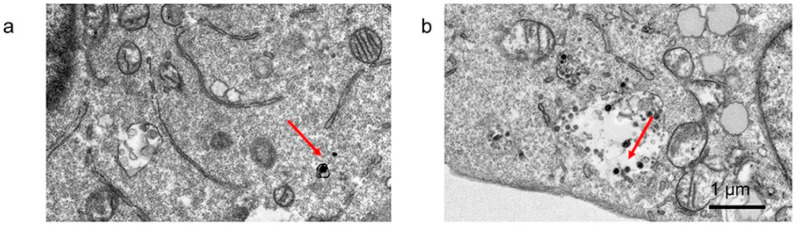
TEM imaging after 4 h of incubation (**a**) with AuPt, and (**b**) with AF. The red arrows indicate the presence of NPs within the cytoplasm of the cells. Reprinted/adapted with permission from [72], 7 May 2025, distributed under a Creative Commons Attribution 3.0 International License.

**Figure 5 cancers-17-02295-f005:**
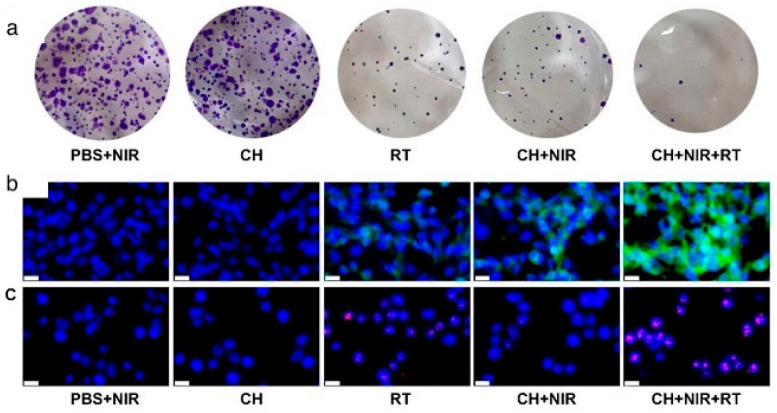
(**a**) Colony formation of 4T1 cells under different treatment conditions. Surviving colonies are stained **purple** with crystal violet; reduced colony number and size in the CH+NIR+RT group indicate enhanced therapeutic efficacy due to synergistic effects (**b**) CLSM images of DCFH-DA-stained 4T1 cells. **Blue fluorescence** indicates DAPI-stained nuclei, while **green fluorescence** represents ROS-positive cells (DCF signal). Enhanced green signal in the CH+NIR+RT group denotes elevated ROS generation. Scale bars: 20 μm (**c**) immunofluorescence staining with γH2AΧ for DNA damage detection. **Blue fluorescence** shows nuclear staining (DAPI), and **red fluorescence** indicates γH2AΧ-positive DNA damage foci. Scale bars: 20 μm. Reprinted/adapted with permission from [82], 7 May 2025, distributed under a Creative Commons Attribution 4.0 International License.

**Figure 6 cancers-17-02295-f006:**
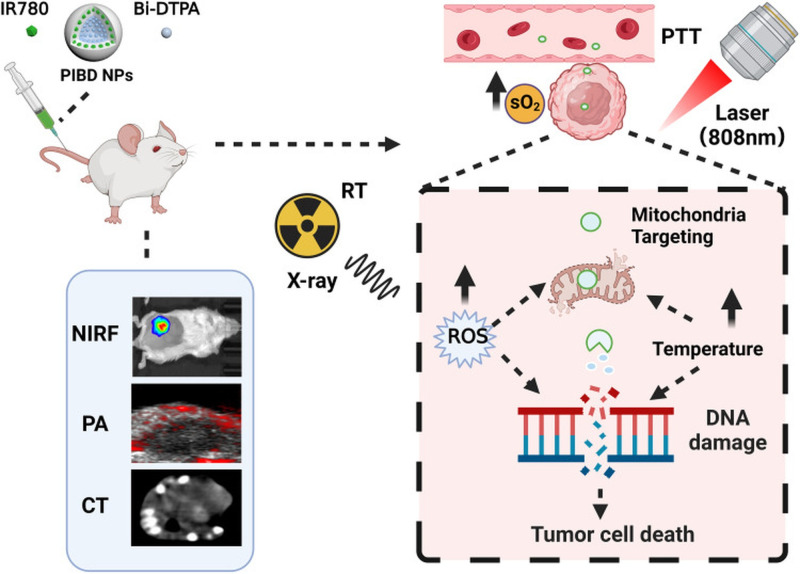
PIBD preparation and mechanism of action for RT +PPT + multimodal imaging. Reprinted/adapted with permission from [91], 7 May 2025, distributed under a Creative Commons Attribution 4.0 International License.

**Figure 7 cancers-17-02295-f007:**
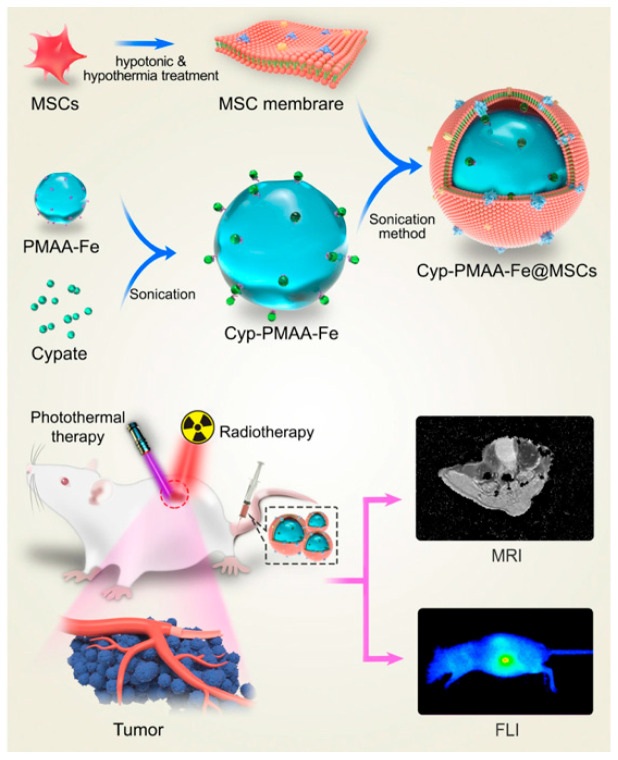
Cyp-PMAA-Fe@MSC NP formation for NSCLC MR and FL image-guided PPT + RT. Reprinted/adapted with permission from [97], 7 May 2025, distributed under a Creative Commons Attribution 4.0 International License.

**Table 2 cancers-17-02295-t002:** Studies conducted with Pt-based NPs.

NP Type	NP Size (nm)	In Vivo/In Vitro	Concentration	Laser Conditions	X-Ray Conditions	Results	Citation
PtNPs	~12	In vitro (B16/F10 melanoma cells)	10–250 μg/m	808 nm, 1.0–1.5 W/cm^2^, 10 min	2–6 Gy	10% viability (triple combo)	[76]
AuPt (AF) NPs	16	In vitro (4T1 breast cancer cells) and in vivo (BALB/c mice)	0–200 μg/m	808 nm, 1 W/cm^2^, 5 min	6 Gy	94% tumor inhibition	[72]
PEG-Au@Pt nanodendrites	30	In vitro (4T1)	0–75 μg/mL	808 nm, 1 W/cm^2^, 10 min	4 Gy	30% viability (triple combo)	[77]

**Table 3 cancers-17-02295-t003:** Studies conducted with Cu-based NPs.

NP Type	NP Size (nm)	In Vivo/In Vitro	Concentration	Laser Conditions	X-Ray Conditions	Results	Citation
CuS NPs w/platelet coating	100 nm	In vitro (CT26 colon cancer cells) and in vivo (CT26 xenograft in mice)	200 ppm	808 nm, 1 W/cm^2^	4 Gy	>80% tumor reduction	[81]
CuP in CH hydrogel	100 nm	In vitro (4T1 breast cancer cells) and in vivo (4T1 xenograft in mice)	0–80 μg/mL	1064-nm laser 0.5 W/cm^2^, 5 min	4 Gy	10% viability	[82]
PEG-[^64^Cu]CuS NPs	11.9 nm (TEM), 29.8 nm (DLS)	In vivo (ATC)	20 µL, OD8	980 nm, 2.5 W/cm^2^, 2 min	7.4 MBq/mouse	83.14% tumor inhibition	[83]

**Table 4 cancers-17-02295-t004:** Studies conducted with Bi-based NPs.

NP Type	NP Size (nm)	In Vivo/In Vitro	Concentration	Laser Conditions	X-Ray Conditions	Results	Citation
Bi@RBC-FA	10–70 nm	In vitro (4T1 breast cancer cells) and in vivo (4T1 xenograft in mice)	100 μg/mL	808 nm, 0.75–2 W/cm^2^	4 Gy	6× ROS increase	[90]
PIBD (PLGA@IR780-Bi)	~100	In vitro (4T1 breast cancer cells) and in vivo (4T1 xenograft in mice)	0–12.5 μg/mL	808 nm, 1 W/cm^2^, 5 min	4Gy	3.2% viability, CI = 0.662	[91]
Porous Bi-PVP NPs	~50 nm	In vivo (HeLa)	n/a	808 nm, 1 W/cm^2^, 5 min	0–8 Gy	Complete inhibition	[93]

**Table 5 cancers-17-02295-t005:** Magnetic and multimodal nanoparticles.

NP Type	NP Size (nm)	In Vivo/In Vitro	Concentration	Laser Conditions	X-Ray Conditions	Results	Citation
GO-SPIO-Au NFs	Varies	In vitro (CT26 colorectal adenocarcinoma) and in vivo (CT26 xenograft mice)	50 µg/mL	808 nm, 1.8 W/cm^2^, 5 min	6 Gy	Viability: NF + RT + PT 18%, RT: 45%, PTT: 50%; almost complete tumor regression in vivo	[95]
INS NPs (FeCo + IR-780)	~40 nm	In vitro (CT26 colorectal adenocarcinoma) and in vivo (CT26 xenograft mice)	200 µg/mL	808 nm, N/A	4 Gy	60% tumor eradication; increased retention and uptake; minimal toxicity	[96]
Cyp-PMAA-Fe@MSCs	248.4 nm	In vivo (NSCLC xenograft mice)	0–200 µg/mL	808 nm, 1.5 W/cm^2^, 5 min	0–8 Gy	RT + PTT + NP: 37% tumor reduction; RT and PTT alone caused tumor growth	[97]
GdW@BSA NCs	~5 nm	In vitro (HeLa cancer cells) and in vivo (MDA-MB-231 xenograft mice)	1 mg/mL	808 nm, 1 W/cm^2^	6 Gy	Cell survival: RT + PTT: 5%; PTT alone: 10%; RT alone: 60%	[100]

**Table 6 cancers-17-02295-t006:** Studies conducted with other metallic NPs.

NP Type	NP Size (nm)	In Vivo/In Vitro	Concentration	Laser Conditions	X-Ray Conditions	Results	Citation
Fe_3_O_4_@Au/rGO	10–60 nm	KB oral squamous carcinoma (In vitro)	20 μg/mL	NIR, not specified	2 and 4 Gy	Cell viability reduced to 11.9% (combined); RT alone: 50.2%, PTT alone: 27%	[73]
ZrC NPs	~100 nm	In vitro (4T1 breast cancer cells) and in vivo (4T1 xenograft in mice)	250 µg/mL	808 nm, 2.0 W/cm^2^	4 Gy	Viability 5% (combined); RT + ZrC: 38.7%, PTT + ZrC: 39%; RT alone: not specified	[101]
RGD-PEG-PAA-MN@LM	~167 nm	In vitro (HepG2 liver cancer cells) and in vivo (HepG2 xenograft in mice)	50–800 µg/mL	808 nm, 2.0 W/cm^2^, 6 min	6 Gy	Tumor volume reduced to 12.6 mm^3^ vs. 790 mm^3^ (control)	[102]
WS_2_ QDs	~3 nm	In vitro (4T1 breast cancer cells) and in vivo (BEL-7402 liver cancer xenograft in mice)	100 µg/mL	808 nm, 1 W/cm^2^, 10 min	4 Gy	Cell survival 6% (combined); RT alone: 75%; PTT: 39%; RT + QDs: 31%	[103]

**Table 7 cancers-17-02295-t007:** Nanoparticle types: advantages and disadvantages for RT + PTT.

NP Type	Advantages	Disadvantages	Ref.
GNPs	Strong LSPR, excellent PTT efficiency, good radiosensitization, easy surface modification	Non-biodegradable, potential long-term accumulation, rapid clearance, expensive	[116,117]
SPIONs	MRI contrast agent, magnetic targeting, biocompatible	Low PTT efficiency, oxidative instability, aggregation in biological media	[118,119]
Graphene-based	High surface area, good PTT efficiency, multifunctional drug loading	Questionable biodegradability, potential inflammation/toxicity, batch variability	[120,121]
Bi-based	Strong X-ray absorption, moderate PTT effect, radiosensitization	Heavy metal toxicity risk, potential bioaccumulation, synthesis complexity	[115,122]
W-based	High Z, good PTT performance, dual-modal imaging (CT/MRI)	Renal clearance issues, potential long-term retention	[103]
Gd-based	Excellent MRI contrast, dual CT/MRI imaging, good radiosensitization	Long-term safety of Gd, retention risk if not ultrasmall, synthesis cost	[123]
Cu-based	High PTT efficiency, ROS generation via Fenton-like reactions, cost-effective	Instability, uncontrolled ion release, cytotoxicity at high doses	[114,124]
Pt-based	Strong radiosensitization, catalytic ROS generation, PTT effect	Non-biodegradable, nephrotoxicity potential, complex synthesis	[125]

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
