# Peer review of "Recent Progress of Nanomedicine for the Synergetic Treatment of Radiotherapy (RT) and Photothermal Treatment (PTT)"

_cancers, 2025, doi:10.3390/cancers17142295_

Round 1
Reviewer 1 Report
Comments and Suggestions for Authors
The review is comprehensive, but it contains minor errors, such as missing spaces, faulty subscripts and superscripts, failing to introduce abbreviations, using incorrect units, and providing an incorrect link to one figure. Someone far better than me at finding these types of errors should review it.
The abstract and references are OK
Details are in the attachment.

Reviewer 2 Report
Comments and Suggestions for Authors
The authors presented the paper "Recent progress of nanomedicine for the synergetic treatment of radiotherapy (RT) and photothermal treatment (PTT)." I have some major comments.
1) The introduction section is very short. I recommend inserting a) advantages of nanoparticles for cancer treatment (e.g., selective delivery by the magnetic field, magnetic hyperthermia, pH-sensitive release, good capacity for drug loading, etc.), and maybe diagnosis (e.g., MRI); b) a much better explanation should be inserted why it is the synergetic treatment of radiotherapy, and photothermal treatment is a focus of the review. c) Why is only this synergetic treatment, according to the author's opinion, the best? Why aren't chemotherapy and radiotherapy?
2) The conclusion section is poor. The novelty of the work should be highlighted. The future outlook should be proven and discussed in detail. The listing of "multimodal imaging, immunotherapeutic integration, and personalized nanomedicine" is not a good future outlook. It can be easily found that many papers provide a lot of information about the topic of nanoparticles.
3) Lines 50-60. Please specify the correct type of nanoparticles for your application. What metal, in what form (bulk metal, oxide, maybe, nanodots, etc.), what sizes, etc?
The same goes for lines 80-82.
There are many known metals and nanoparticles, but not all of them have such properties.
4) It will be excellent to provide the same Table as Table 1 for all other nanoparticles in sections 1.5-1.9.
5) Discussion and Future Perspectives or Limitations. I recommend inserting a Table with different types of nanoparticles and their advantages and disadvantages.
Reviewer 3 Report
Comments and Suggestions for Authors
The present review presents the most recent literature in NP-mediated RT–PTT combination therapies, emphasizing on the dual-functional role of NPs in enhancing radiation-induced cell damage while simultaneously promoting photothermal conversion efficiency. The research direction is important, but I have some specific comments:
-Please clarify the term «nano-oncology» and potencial of NP-mediated RT–PTT combination therapies for nano-oncology.
-The abstract should clearly highlight the novelty of the methodology or findings and reflect a central theme of the article.
- Authors should compare the NP-mediated RT–PTT combination therapies with some another oncological treatments.
- Why there is no differentiation of oncological diseases. The choice of treatment method depends on this.
- Could the authors provide more details on the potential limitations of NP-mediated RT–PTT combination therapies and risks of this treatment?
- There is no analysis and importance of the review for Readers. Please correct it.
- The manuscript ends with an additional discussion that looks like a Introduction section. Please re-write discussion.
- In Discussion section Authors wrote “The purpose of this review is to discuss the most recent advances “ - why it in the end of the manuscript?
- “in vivo” should be in Italic.
Round 2
Reviewer 2 Report
Comments and Suggestions for Authors
Thank you for the revised paper.
Reviewer 3 Report
Comments and Suggestions for Authors
All the queries has answered.